# Treatment Combining CD200 Immune Checkpoint Inhibitor and Tumor-Lysate Vaccination after Surgery for Pet Dogs with High-Grade Glioma

**DOI:** 10.3390/cancers11020137

**Published:** 2019-01-24

**Authors:** Michael R. Olin, Elisabet Ampudia-Mesias, Christopher A. Pennell, Aaron Sarver, Clark C. Chen, Christopher L. Moertel, Matthew A. Hunt, G. Elizabeth Pluhar

**Affiliations:** 1Department of Pediatrics, University of Minnesota, Minneapolis, MN 55455, USA; olin0012@umn.edu (M.R.O); ampud001@umn.edu (E.A.-M.); moert001@umn.edu (C.L.M.); 2Masonic Cancer Center, University of Minnesota, Minneapolis, MN 55455, USA; penne001@umn.edu (C.A.P.); sarver@umn.edu (A.S.); 3Department of Laboratory Medicine and Pathology, University of Minnesota, Minneapolis, MN 55455, USA; 4Institute for Health Informatics, University of Minnesota, Minneapolis, MN 55455, USA; 5Department of Neurosurgery, University of Minnesota, Minneapolis, MN 55455, USA; ccchen@umn.edu (C.C.C.); huntx188@umn.edu (M.A.H.); 6Department of Veterinary Clinical Sciences, University of Minnesota, St. Paul, MN 55108, USA;

**Keywords:** glioma, immunotherapy, tumor lysate, dogs

## Abstract

Recent advances in immunotherapy have included inhibition of immune checkpoint proteins in the tumor microenvironment and tumor lysate-based vaccination strategies. We combined these approaches in pet dogs with high-grade glioma. Administration of a synthetic peptide targeting the immune checkpoint protein, CD200, enhanced the capacity of antigen-presenting cells to prime T-cells to mediate an anti-glioma response. We found that in canine spontaneous gliomas, local injection of a canine-specific, CD200-directed peptide before subcutaneous delivery of an autologous tumor lysate vaccine prolonged survival relative to a historical control treated with autologous tumor lysate alone (median survivals of 12.7 months and 6.36 months, respectively). Antigen-presenting cells and T-lymphocytes primed with this peptide suppressed their expression of the inhibitory CD200 receptor, thereby enhancing their ability to initiate immune reactions in a glioblastoma microenvironment replete with the immunosuppressive CD200 protein. These results support consideration of a CD200 ligand as a novel glioblastoma immunotherapeutic agent.

## 1. Introduction

Immunotherapy is the third important wave in the history of cancer treatment. Clinical success of immune checkpoint blockade with monoclonal antibodies has led to an estimated 3000 immuno-oncology trials [1]. However, the pertinence of immune checkpoints in tumor vaccine-based therapeutic strategies has received less attention. In principle, the presence of immune checkpoint molecules in tumor vaccines would compromise the ability of the local antigen-presenting cells (APCs) to activate an adaptive immune response against tumor-specific antigens, thereby compromising therapeutic efficacy.

During an adaptive immune response, expression of certain cell-surface proteins downregulates or terminates immune activation [2]. The observation that these proteins, known as immune checkpoints, are expressed in solid tumors was a seminal discovery in immuno-oncology [3]. These molecules on tumor cells interact with their cognate receptors on T-lymphocytes to suppress the signaling required for T-cell activation upon antigen presentation, thereby allowing the tumor to evade immune surveillance. These findings underlie efforts to develop checkpoint inhibitors to reverse immunosuppression systemically as well as in the tumor microenvironment.

There has been focused attention recently on dogs as a translational model due to their strong anatomical and physiological similarities to humans and the sheer number of pet dogs that are diagnosed and managed with cancer each year [4,5,6]. Dog owners are highly motivated to seek out new treatment options for their pets with cancer and will readily participate in a clinical trial if conventional therapy does not meet their goals. Strong similarities have recently been shown between the canine and human genome, especially with respect to gene families associated with cancer. These combined factors suggest cancer in companion canines are a viable model for pre-clinical human cancer research including brain tumors [7,8,9]. The size of dogs makes multimodality protocols involving surgical interventions more feasible than murine models. Furthermore, the lack of “gold standard” treatments permits early and humane testing of novel therapies.

Here we examine whether an inhibitor of the CD200 (OX2) immune checkpoint augments the efficacy of autologous tumor lysate vaccines against high-grade glioma, including glioblastoma. Glioblastoma is the most common form of primary adult brain cancer [10]. The prognosis of patients afflicted with glioblastoma remains dismal with a median survival of 14.6 months despite aggressive surgical resection, chemotherapy, and radiation therapy [11]. Although the central nervous system (CNS) was once thought to be an immune-privileged site [12], subsequent studies have revealed that both innate and adaptive immune systems play key roles in the host response in glioblastoma pathogenesis [13]. This knowledge has led to investigation of tumor vaccines as a therapeutic modality for glioblastoma. In this approach, autologous antigens from resected tumor are injected to elicit an immune response against any residual tumor cells. However, glioblastoma cells express several immune checkpoint molecules as cell surface and soluble proteins [14] that neutralize an immune reaction to the tumor [15]. Our earlier studies using tumor lysate-based vaccines in pet dogs with high-grade glioma after tumor resection showed an increased survival over surgical resection alone from 66 days [16] to 196 days (unpublished data). Although this was a significant improvement in prognosis, it fell far short of the results we anticipated. These results may be explained by the fact that in addition to immunosuppression in the tumor microenvironment, checkpoint molecules in tumor lysate vaccines may suppress anti-tumor immune cell activation. Indeed, we reported CD200 in tumor lysates compromises the ability of local APCs to initiate an adaptive immune response to tumor-specific antigens [17].

CD200 is an immune checkpoint protein related to the B7 family of co-stimulatory receptors required for T-cell activation and signaling [18]. The role of CD200 as an immune checkpoint protein was substantiated by the finding that CD200-deficient mice exhibit auto-immune phenotypes [19]. Importantly, CD200 is expressed in a wide spectrum of cancers, including chronic lymphocytic leukemia (CLL) [20], multiple myeloma [21], acute myeloid leukemia [22], melanoma [23], ovarian tumors [24], metastatic small cell carcinoma [25], and glioblastoma [10]. CD200 is expressed on the surface of tumor cells and can be released in a soluble form when cleaved by metalloproteases such as ADAM 28, A Disintegrin, and Metalloprotease 28 [26]. Notably, plasma levels of soluble CD200 correlate with tumor burden and survival in CLL patients [27]. The physical interaction between cancer-released CD200 and its inhibitory receptor (CD200R1) on APCs (Figure 1) suppresses secretion of pro-inflammatory cytokines, including interleukin 2 (IL2) and interferon gamma (IFNγ) [17,28] increases production of myeloid-derived suppressor cells (MDSCs) [10] and regulatory T-cells (Tregs) [29,30]; and compromises an anti-tumor immune response.

We previously demonstrated that the CD200–CD200R1 interaction is central to maintaining the glioblastoma immunosuppressive microenvironment and developed peptides targeting these interactions [10]. Subsequent studies revealed that these peptides mediate their effects through separate CD200 activation receptors (CD200ARs), rather than through the inhibitory CD200R1 (Figure 1). Moreover, our previous murine experiments [10] suggest that immuno-stimulation can be achieved by administering peptides of CD200 as ligands (CD200AR-L) that specifically target CD200ARs. The CD200AR-L only provided survival benefits in our murine glioma model when administered in combination with tumor lysate vaccines. The exact sequence of the peptide was shown to be vital when scrambled peptides given with tumor lysate failed to stimulate an anti-tumor response in our murine glioma model [10]. Immune activation of human dendritic cells was demonstrated after stimulation with a human-specific CD200AR-L, but not when the cells were pulsed with a scrambled peptide (unpublished data). We found the therapeutic effects of these CD200 peptides compelling and wished to translate these findings into the clinical setting. Because of sequence divergence between the murine and canine CD200AR-L receptors, we developed canine-specific CD200AR-Ls to further assess the efficacy of these peptide ligands. We now show local intradermal injection of the canine CD200AR-L prior to administration of autologous tumor lysate significantly enhances the efficacy of the lysate vaccines in pet dogs with spontaneous high-grade glioma.

## 2. Results

The efficacy of tumor lysate vaccination was augmented by the addition of CD200AR-L peptide in a spontaneous canine glioma model. Our laboratory has a long-standing interest in the development of autologous tumor lysate vaccination as a strategy for treating glioblastoma [31,32]. We postulated that the presence of immune checkpoint proteins in the tumor lysate may compromise the efficacy of this therapy and that treatment with CD200 peptide ligands would modulate the effects of any CD200 checkpoint protein in the lysate vaccine. To test our hypotheses, 20 dogs with spontaneously occurring high-grade glioma definitively diagnosed with histopathology were treated by tumor resection and serial intradermal vaccinations of canine CD200AR-L + autologous tumor lysate and topical imiquimod at the sites of injection (Appendix A). Tumor progression-free and overall survival times were compared with historical data from a cohort of dogs previously treated with autologous tumor lysate vaccines alone after tumor resection. The extent of tumor resection and type of high-grade glioma for these two cohorts were comparable. We observed a two-year progression-free survival rate of 30% in dogs that received the canine CD200AR-L (Figure 2). Death was attributed to tumor recurrence in 60% (12/20) of the dogs, 25% (5/20) of the dogs died or were euthanized and had no evidence of disease on postmortem examination of their brains, and three dogs are still alive at the time of this writing between 780 and 890 days after surgery. In contrast, tumor progression was the cause of death in 87% (13/15) of the dogs treated with tumor lysate alone after surgery. The median overall survival of dogs treated with canine CD200AR-L administration before autologous-tumor-lysate inoculation was 12.9 months. This survival time compared favorably to the 6.83-month survival of dogs treated with tumor lysate vaccination alone (Figure 2). One dog with tumor recurrence 18 months (548 days) after surgery was treated with a second surgery followed by autologous tumor lysate vaccine co-administered with the canine CD200AR-L, leading to an additional 9.93 months of survival, for a total of 28 months. This dog was not included in Figure 2.

Interestingly, five dogs had residual tumor (7–40% of the original volumes) after surgery, but radiologic evidence of tumor regression was seen by magnetic resonance imaging (MRI) four months after co-administration of CD200AR-L and tumor lysate vaccine. We never observed this type of response in dogs treated with tumor lysate vaccine treatment alone after surgery. Five dogs developed cerebral leukoencephalopathy characterized by T2 hyperintensity of the periventricular white matter tracts and ventricular dilatation. However, these radiologic findings resolved following treatment with anti-inflammatory doses of corticosteroids. Vaccinations were discontinued during the corticosteroid therapy in one dog that developed symptoms of CNS disease including hemiparesis and episodes of breakthrough generalized seizures despite chronic anti-epileptic drug (AED) administration and, although the T2 hyperintensity resolved, tumor recurrence was noted on an MRI performed 2 months later. Immunotherapy was reinitiated when the dog recovered, and tumor regression was noted after two rounds of tumor lysate and CD200AR-L injections.

Because serum soluble CD200 (sCD200) levels correlated with tumor burden and overall survival in human ependymoma patients [10], we measured serum concentrations of CD200 in the canine patients; they appeared to be predictive of tumor progression in at least one case (Figure 3A–3C). There was no evidence of treatment-related adverse effects based on blood tests, physical and neurological examinations, and post-mortem examination. These data suggest the potential utility of serum CD200 as a companion biomarker for CD200AR-L therapeutic strategies.

## 3. Discussion

The present study provides evidence of the efficacy of immune checkpoint inhibition at the site of autologous tumor vaccination to provide prolonged progression-free and overall survival times in a large animal model of spontaneous glioma. We based this work on evidence that survival of human glioblastoma (GBM) patients is correlated with the expression of CD200/CD200R1-related genes. We analyzed gene expression profiles of human GBM tumor samples in The Cancer Genome Atlas (TCGA) dataset using Gene Cluster Expression Summary Score (GCESS) [33]. We identified gene clusters that are concordantly expressed across the dataset and associated with overall survival in an unbiased statistical analysis. CD200R1 expression was found within a large cluster of genes highly enriched in immune-related transcripts. Increased transcript levels of the genes in this cluster were significantly associated with decreased survival times (Figure 4A, Table 1). Patients whose tumors expressed high levels of the CD200R1 containing cluster (Cluster 14) had shorter overall survival times compared to those with tumors that expressed lower levels of the cluster (Figure 4B–4D). These results suggest the critical importance of the CD200/CD200R1 interaction to mediate an immunosuppressive microenvironment in GBM.

The data from the canine clinical trial are consistent with our previous studies [10,17], which provided compelling evidence that the presence of the immune checkpoint protein, CD200, in tumor lysates suppressed the ability of local APCs to activate and recruit T-cells to trigger an effective anti-tumor immune response. We built on this observation and tested whether a peptide ligand of CD200 could modulate CD200-mediated immunosuppression. Our results indicated that murine CD200AR-L enhanced the ability of APCs to induce an antigen-specific response. Of note, murine CD200AR-L priming prior to antigen presentation to the T-cells suppressed expression of inhibitory receptor, CD200R1, rendering these cells resistant to the effects of CD200 in the glioblastoma microenvironment. Local intradermal injection of these immune-stimulatory molecules minimizes the likelihood of the toxicities associated with systemic administration of immune modulators.

Although murine brain tumor models have yielded valuable insights into the etiology of glioblastomas, therapies that showed enormous promise in these models have frequently failed in clinical translation. Expert panels assembled by the National Institutes of Health to address this issue have failed to come to a consensus about an optimal therapeutic model. In the absence of such consensus, therapeutic testing in animals other than mice warrants consideration before designing clinical trials. Our studies in pet dogs with spontaneous disease are particularly relevant in this context. High-grade glioma arises sporadically in certain brachycephalic breeds of dogs including boxers, Boston terriers, and French and English bulldogs [34]. This spontaneous canine “model” is an attractive platform to characterize the immunologic effects of CD200 because (1) the tumors exhibit histologic features highly similar to those observed in human glioblastoma; (2) the tumors arise and grow in an immune microenvironment free from experimental manipulation; (3) there are substantial similarities in terms of immune cell populations in histopathologic samples of human and canine glioma; (4) high-grade glioma are typically diagnosed at a late stage in dogs when tumor burden results in severe neurologic deficits; and (5) significant homology exists between human and canine CD200. When taken in this context, the efficacy of canine CD200AR-L in the canine model provides strong support for future human clinical trials.

In current clinical practice, response to therapy is primarily assessed using MRI to measure tumor volume. This practice is problematic because MRIs are costly, are incapable of allowing distinction between true disease progression and pseudo-progression, and have insufficient resolution to detect tumor growth at the microscopic level. Therefore, there is a critical need for minimally invasive biomarkers or liquid biopsies to assess disease state in glioblastoma patients. In many aspects, the biologic properties of CD200 render it well suited as a biomarker for tumor burden. Although CD200 is a cell-surface glycoprotein, much of the CD200 in glioblastoma patients is released as a soluble factor [17]. Independent groups have determined that serum levels of soluble CD200 in cancer patients correlates with tumor burden, which was supported by data in this manuscript. In aggregate, these results support serum CD200 as a potential “liquid biopsy” platform for glioblastoma monitoring.

In summary, our study provides data to support the use of peptide ligands that interfere with the biologic function of the CD200 immune checkpoint prior to tumor lysate vaccine administration as a novel therapy for glioblastoma.

## 4. Materials and Methods

### 4.1. Peptide Synthesis

Peptides were designed and subsequently manufactured (Thermo Fisher Scientific, Rockford, IL) to represent (1) regions of the CD200 protein previously shown to interact with CD200AR and (2) sequences with the greatest homology among murine, canine, and human CD200 molecules [35]. Murine and human APCs pulsed with these peptides were shown to produce high levels of inflammatory cytokines [10]. The purity of the peptides was >95%, and each peptide was modified by N-terminal acetylation and C-terminal amidation to enhance its stability.

### 4.2. TCGA Analysis

The Cancer Genome Atlas (TCGA) data portal (http://tcga-data.nci.nih.gov/tcga) was used to download survival times, death events, and RNAseq data for human patients with glioblastoma. Cluster 3.0 (C Clustering Library 1.52) was used to log_2_ transform and mean-center the gene expression data and then to perform hierarchical average linkage clustering using the Pearson similarity metric on all genes with standard deviation >1.0. Gene clusters with a dendrogram node correlation >0.20 and ≥50 individual genes were identified and associations with outcome were generated as previously described [30]. (Figure 4A, Table 1) Associations between the Gene Cluster Expression Summary Scores (GCESS) for CD200R1-containing cluster and outcomes are reported using Kaplan–Meier analyses of groups generated using quartiles of GCESS values. (Figure 4B–D) The list of genes co-localized to cluster 14 is included as CD200R1_cluster_gene list.xls.

### 4.3. Canine Study

Pet dogs with a solitary intra-axial mass suspected to be a glioma found by MRI were recruited into a pilot study to assess the effect of a canine-specific peptide on vaccine-based immunotherapy using a protocol approved by the Institutional Animal Care and Use Committee. The MRIs were performed to find the cause of sudden behavioral changes or generalized seizures in mature dogs that were treated with AEDs and corticosteroids to control seizures and minimize cerebral edema, respectively. Additional inclusion criteria were (1) tumor suitable for surgical resection, (2) informed consent provided by the dog’s owner, (3) normal mentation status at the time of surgery, and (4) no previous definitive therapy for the suspected glioma.

All dogs underwent surgical resection of the intra-axial mass for gross resection of the tumor, to reduce compression and intra-cranial pressure, and to obtain tissue for definitive histopathology and vaccine preparation. An immediate postoperative MRI was performed to assess extent of resection and measure residual tumor volume. Tumor volume was measured using the planimetry method where tumor area is calculated using Osirix software after outlining the tumor perimeter on each MR slice. Tumor volume equals the sum of slice areas multiplied by slice thickness plus gap width. After recovery from anesthesia, all dogs were monitored in the intensive care unit for at least 24 h. Anti-epileptic drugs (phenobarbital, levetiracetam, and zonisamide; alone or in combination) were continued after surgery, and the corticosteroid dose was tapered and discontinued within 10 to 14 days.

Tumor cell lysates were prepared by culturing single-cell suspensions from minced fresh tumor samples at 37 °C in 5% O_2_ as described [31]. Cultured tumor cells were lysed by multiple freeze-thaw cycles and then irradiated (20 Gy). The first vaccination was given 10 days after surgery and was repeated weekly for 3 doses, then once every 4 weeks for 3 doses, and then every 6–8 weeks until tumor progression or death (Appendix A). The vaccine protocol was as follows: the canine-specific CD200AR-L peptide (5 μg/kg) was injected intradermally on the nape of the neck. Twenty-four hours later, imiquimod (1 packet 5% cream/12.5 g) was applied topically to the skin and allowed to absorb for 10–15 min prior to intradermal injection of autologous tumor lysate (~500 μg of protein) mixed with CD200AR-L peptide (5 μg/kg). Disease status was monitored by MRI and physical examination at 4, 8, and 12 months after surgery, and then as needed if tumor recurrence was suspected.

Surgical resection was defined as gross total resection if there was no residual contrast enhancement for gadolinium-enhancing lesions or T2-weighted FLAIR (fluid-attenuated inversion recovery) hyperintensity for non-enhancing lesions, subtotal resection if <10% of the original tumor volume remained, or partial resection if there was >10% residual tumor volume. Clinical response on follow-up MRIs was considered a complete response (CR) if there was no evidence of the tumor; partial response (PR) if the tumor volume had decreased by ≥65%; progressive disease (PD) if the tumor volume increased ≥40%; or stable disease if the response did not qualify as CR, PR, or PD as defined above. An MRI was performed if a dog developed recurrent or worsening neurologic signs before a scheduled MRI.

## 5. Conclusions

Glioblastoma is the most common and deadly primary intra-axial CNS tumor in people and dogs. Numerous ongoing clinical trials targeting immune checkpoints have failed to enhance survival in patients with CNS tumors. We targeted the CD200 checkpoint which controls the immune system through paired inhibitory and activation receptors. In this study, we demonstrated that certain CD200 peptides enhance the efficacy of autologous tumor cell lysate therapeutic vaccines in pet dogs with high-grade glioma as evidenced by significantly increased survival times. This strategy may have similar efficacy when translated to glioblastoma patients.

## Figures and Tables

**Figure 1 cancers-11-00137-f001:**
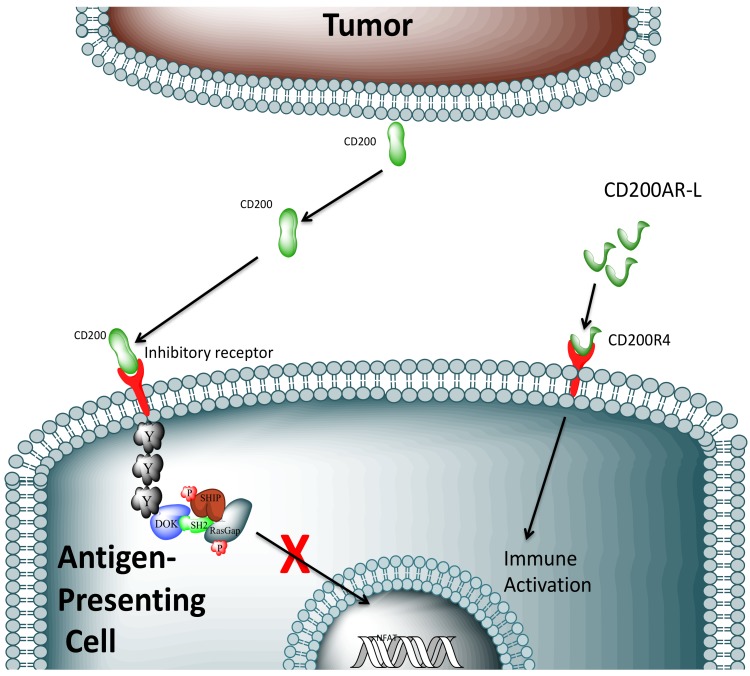
Experimental model. Full-length CD200 secreted from tumor cells binds to the CD200 inhibitory receptor (CD200R1) on antigen-presenting cells, shutting down an immune response. In addition to the inhibitory receptor, the CD200 checkpoint has CD200-like activation receptors (CD200R4) that have adjuvant-like properties when stimulated; ligation with the CD200 activation receptor peptide ligand (CD200AR-L) activates an as yet undetermined signaling cascade that surmounts the inhibitory signals and allows activation of and normal antigen presentation by the antigen presenting cell. Permission by Michael R. Olin, University of Minnesota, Minneapolis, MN. Unpublished work, 2018.

**Figure 2 cancers-11-00137-f002:**
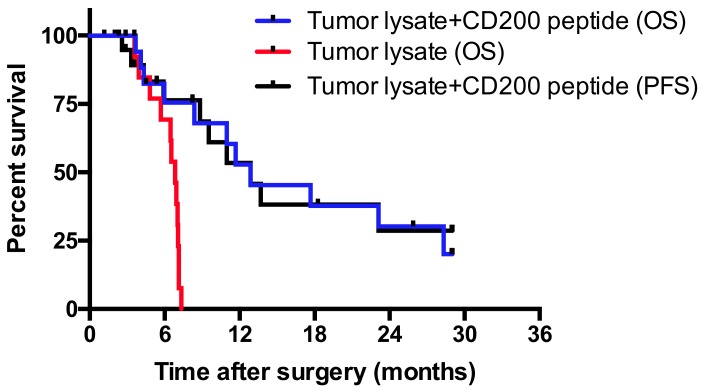
Treatment with the CD200 inhibitor extends progression free and overall survival in dogs. Following surgical resection of the tumors, serial vaccinations of autologous tumor lysate and canine-specific CD200AR-L were administered to each dog. Disease status was followed using magnetic resonance imaging (MRI) and calculated progression-free (black line) and overall survival (blue line) times were significantly longer than a cohort of dogs treated with tumor lysate alone after surgery (red line) (*p* = 0.0001). Ticks represent censored cases because the dog is still alive (*n* = 3) or had no evidence of disease on postmortem examination of the brain (*n* = 6).

**Figure 3 cancers-11-00137-f003:**
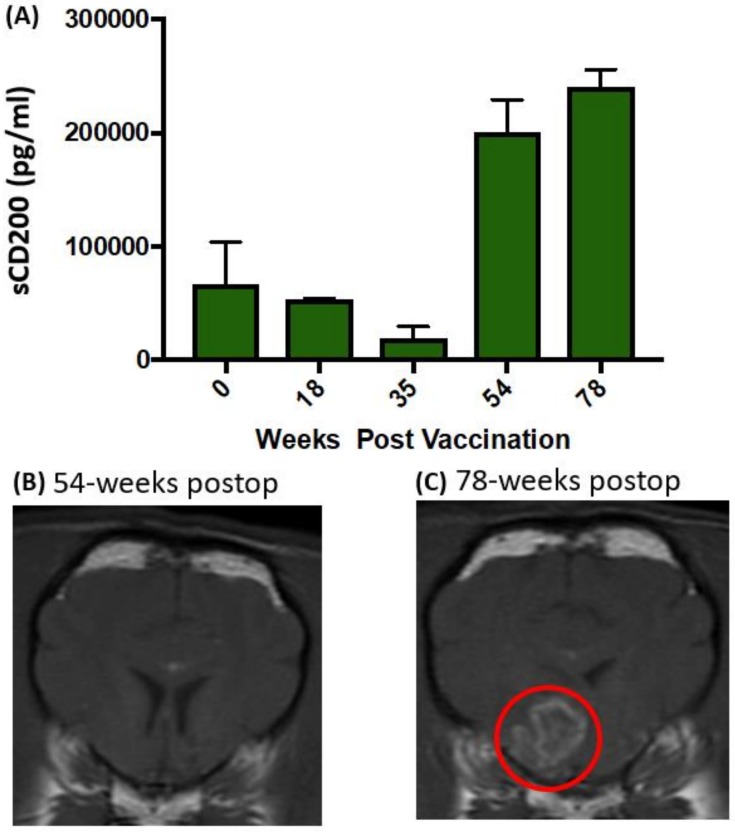
Soluble CD200 (sCD200) predicts tumor recurrence prior to MRI evidence. (**A**) Serum levels of sCD200 decreased after surgery and vaccinations of autologous tumor lysate + CD200AR-L in one Boston terrier with a grade III glioma. (**B**) Serum sCD200 increased at 1 year although there was no evidence of tumor recurrence on the MRI at that time. (**C**) Six months later, an MRI was repeated when the dog developed severe breakthrough generalized seizure activity and tumor progression was seen (red circle).

**Figure 4 cancers-11-00137-f004:**
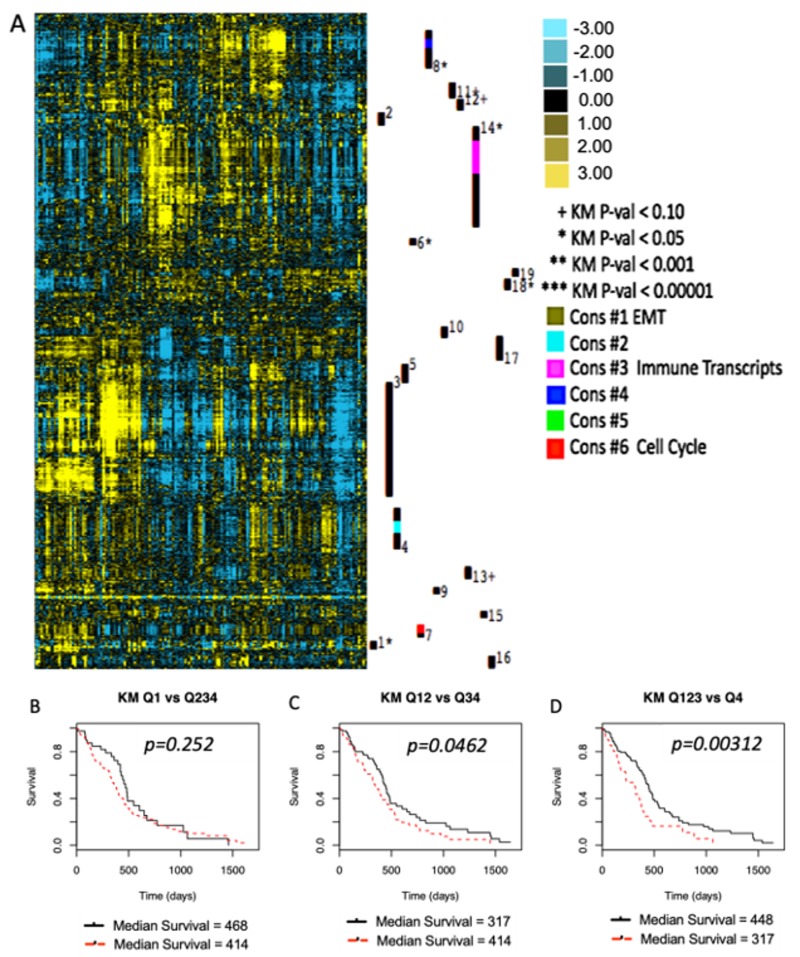
CD200R1-related genes are associated with shorter overall survival in humans. (**A**) Transcriptome profile for glioblastoma showing clusters of genes associated with overall survival were analyzed in patient tumor samples available in The Cancer Genome Atlas database. Transcripts with increased levels are shown in yellow, while transcripts with decreased levels are shown in blue. Transcript level clusters with correlation >0.60 and containing ≥60 genes were systematically identified using Gene Cluster Expression Summary Scores. These clusters are visualized with a numbered black bar to the right of each of the heatmaps. Those with significant differences in survival based on Kaplan–Meier analyses are indicated with symbols and their associated *p*-values are shown. (**B**–**D**) Kaplan–Meier plots showing that patients expressing high levels of the gene cluster containing CD200 (red lines) had shorter overall survival times than those expressing lower levels (black lines). Comparisons are top quartile (Q1) versus three lowest quartiles (Q234), top two quartiles (Q12) versus the lowest two (Q34), and top three quartiles (Q123) versus the lowest (Q4).

**Table 1 cancers-11-00137-t001:** Gene with increased transcript levels in cluster associated with poor survival.

Index	Name	P-Value	Adjusted P-Value	Z-Score	Combined Score
1	Immune System_Homo sapiens_R-HSA-168256	1.116 x10^−49^	8.88x 10^−47^	−2.23	251.61
2	Immunoregulatory interactions between a Lymphoid and non-Lymphoid cell_Homo sapiens_R-HSA-198933	4.226 x 10^−37^	1.682 x 10^−34^	−2.00	167.65
3	Extracellular matrix organization_Homo sapiens_R-HSA-1474244	1.460 x 10^−29^	3.875 x 10^−27^	−2.10	139.12
4	Adaptive Immune System_Homo sapiens_R-HSA-1280218	2.066 x 10^−21^	3.290 x 10^−19^	−2.25	107.12
5	Class A/1 (Rhodopsin-like receptors) Homo sapiens_R-HSA-373076	9.042 x 10^−21^	1.200 x 10^−18^	−2.10	96.97
6	Chemokine receptors bind chemokines_Homo sapiens_R-HSA-380108	2.178 x 10^−22^	4.334 x 10^−20^	−1.93	96.39
7	Cytokine Signallind in Immune system_Homo sapiens_R-HSA-1280215	5.330 x 10^−17^	6.061 x 10^−15^	−2.35	88.11
8	GCPR ligand binding_Homo sapiens_R-HSA-500792	1.172 x 10^−15^	1.166 x 10^−13^	−2.20	75.57
9	Innate Immune System_Homo sapiens_R-HSA-168249	1.917 x 10^−13^	1.272 x 10^−11^	−2.33	68.28
10	Peptide ligand-binding receptors_Homo sapiens_R-HSA-375276	1.771 x 10^−15^	1.566 x 10^−13^	−1.91	64.80

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
