# Peer review of "Treatment Combining CD200 Immune Checkpoint Inhibitor and Tumor-Lysate Vaccination after Surgery for Pet Dogs with High-Grade Glioma"

_cancers, 2019, doi:10.3390/cancers11020137_

Reviewer 1 Report

This excellent work shows that a combination iof CD200 checkpoint inhibition and tumor lysate vaccination had a strong effect in canine high grade glioma. As high grade gliomas occur spontaneously in dogs, this model has advantages as compared to the widely used murine models. Tumor lysates combined with CD200 checkpoint inhibition strongly prolonged survival as compared to tumor lysates alone. All results are highly convincing, and the conclusion drawn are comprehensible. This work is a sufficient base for initiating a phase I clinical study with glioblastoma patients. Therefore this manuscript will find many interested readers. I strongly recommend publication of this work in Cancers.

I have two suggestions for improvement:

Is there data available of the effect of CD200 checkpoint inhibition alone in dog high grade glioma? Do you suspect that the effect seen has to be ascribed to the combination of tumor lysate and CD200 checkpoint inhibition or was it rather the CD200 checkpoint inhibition alone?

Please discuss that the results of treatment with tumor lysates alone (without checkpoint inhibition) were rather disappointing in clinical studies.

Author Response

Reviewer 1

1.     Is there data available of the effect of CD200 checkpoint inhibition alone in dog high grade glioma? No, we addressed this treatment group in our murine glioma model and treatment with the inhibitor without tumor lysate did not change survival times over the saline control. We think that this lack of effect is due to the low immunogenicity of tumors in situ, and that tumor lysates expose more antigens to provide an antigen specific response. Therefore, we have no reason to believe CD200 inhibition alone would have any therapeutic effect in dogs with high grade glioma, and thought it would not be ethical to include a cohort of tumor-bearing dogs treated just with the inhibitor.

2.     Do you suspect that the effect seen has to be ascribed to the combination of tumor lysate and CD200 checkpoint inhibition or was it rather the CD200 checkpoint inhibition alone? As noted in the response to comment/question #1, we suspect it was the combination. Our model is that tumor antigens in the lysate are taken up and presented to effector T-cells by antigen presenting cells and that CD200 inhibition prevents suppression of effector function.

3.     Please discuss that the results of treatment with tumor lysates alone (without checkpoint inhibition) were rather disappointing in clinical studies. This discussion is now added.

Reviewer 2 Report

The authors report the inhibition of CD200 signaling as an immune checkpoint dramatically ameliorates vaccination therapy of glioma, by using spontaneous glioma in pet dog for the proof-of-concept. This argument has been done consistently throughout their recent works, and since the previous one was performed in the experimental mouse model, current manuscript is relatively clinical despite being a study using animals. 

It is needless to refer to the Novel Prize 2018, since focusing on immune checkpoint is a big biological issue, this manuscript has certain biological as well as medical significance. However, several data and discussion are not enough to properly prove their intention, and moreover, a big strange point is on the way of presentation of data. I point them as follows.

1. An experiment using negative control peptide by which proving, the activation of CD200AR is necessary in boosting the vaccination, is missing.

2. An experiment using positive control peptide or molecule to activate CD200R1 by which proving, the activation of inhibitory role of CD200R1 is antagonistic to immune reaction in the vaccination, is missing.

3. The immuno-stimulative role of a spliced variant form of CD200 which is truncated in N - terminus has already reported (Neoplasia, 18, 229-241, 2016.) Relationship between P4A10 and this truncated form of CD200 should be discussed. 

4. There are no mention in Figure 4 even not only in Results section but also in Discussion section. What is Figure 4 for ?.

Unless experimentally impossible, #1 and #2 should be essential for author's arguments.

On #3, the point should greatly help readers to understand the whole view of CD200 action.

On #4, the way of presentation is so curious to me. I hardly saw this kind of presentation. I feel that building of the logic through the manuscript should be reconsidered.

Author Response

Reviewer 2

General comment: We do not understand why the reviewer checked the “Not applicable” answer to the question: “Are the results clearly presented?” This is in contrast to Reviewer 1 who noted, “All results are highly convincing, and the conclusion drawn are comprehensible.”

1.     An experiment using negative control peptide by which proving, the activation of CD200AR is necessary in boosting the vaccination, is missing. Inclusion of a negative control peptide would not prove activation of CD200AR is necessary; it would show that the order and nature of the amino acids in the peptide were required to augment the tumor lysate vaccine. While we routinely include such “scrambled” peptides as negative controls in murine models, we deemed it unethical to include a similar cohort in the canine studies after our first few dogs treated with the inhibitor plus tumor lysate showed such positive results.

2.     An experiment using positive control peptide or molecule to activate CD200R1 by which proving, the activation of inhibitory role of CD200R1 is antagonistic to immune reaction in the vaccination, is missing. Similar to our answer to the first comment, such an approach is valid in murine models but is not appropriate in the canine model.

3.     The immuno-stimulative role of a spliced variant form of CD200 which is truncated in N - terminus has already reported (Neoplasia, 18, 229-241, 2016.) Relationship between P4A10 and this truncated form of CD200 should be discussed.  The Neoplasia paper reports that rat glioma cells expressing the truncated form of CD200 on their surface do not grow as readily and are more readily recognized and attacked by immune cells compared to those expressing the full-length CD200 molecule. There are many truncated forms of the full-length CD200 molecule that have been shown to have varying effects on immune cells. We present information on a truncated form that is not bound to tumor cells, but injected into the skin in a soluble form. We choose a specific truncated form that was shown to activate immune cells outside the brain in combination with tumor derived antigens in a murine brain tumor model. We hypothesized that this truncated form binds to one of the CD200R-like receptors (as described by Gorczynski et al., which we refer to as CD200ARs) on immune cells and results in immune activation, and are currently working to test that hypothesis using murine cells. This discussion is now added. Furthermore, the information about the human P4A10 peptide has been removed from this manuscript because that information is presented in depth in a separate paper.

4.     There are no mention in Figure 4 even not only in Results section but also in Discussion section. What is Figure 4 for ?. Please accept our apologies for this error. An incomplete earlier draft of the manuscript was submitted inadvertently. The data shown in Figure 4 are now presented in the Materials and Methods section and discussed in the Discussion section.

Round  2

Reviewer 2 Report

To the general comment

I do not understand why having different opinion against reviewer 1 can be the point of criticism toward reviewer 2 from the authors. In fact, indeed, there was a severe inadequate point on the presentation of Figure 4. Why should I evaluate this as “All results are highly convincing, and the conclusion drawn are comprehensible.” to the question “Are the results clearly presented?” ?

There are several illogical points in the answer to the points 1 and 2.

  (1) The authors do not appreciate the additional assessment by using positive and/or negative control peptides suggested by reviewer 2. Then, how they validate and conclude that CD200AR really works in their vaccination system? Additionally, even though they do not agree to the suggestion, they did that “dispensable” experiments in rodents, and stand on this previously executed study as one of the reason to refuse the suggestion from reviewer 2. Why the study should be done on only rodents? Why the study is dispensable on canine? Do they have any evidence that CD200 action is same between rodents and canine?

(2) As rebuttal to the study using positive and/or negative control, they point an ethical reason. Then, how they explain the pertinence of the placebo experiment in clinical research in human patients.

(3) What is the ethical reason why placebo experiments can be done on human but control experiments can not be done on canine? Similarly, what is the ethical reason why control experiments can be done on rodents but not in canine?

3. Good discussion. Hope good results for the current work.

4. To whom the apology from the authors was made? If to the reviewer 2, the reviewer will willing to accept the apology, however, are there any scientific meanings/significance? Should the apology be made to a reviewer, and be included in scientific comments? I would like to ask the decision of the editor.

Author Response

 Comments and Suggestions for Authors

To the general comment

I do not understand why having different opinion against reviewer 1 can be the point of criticism toward reviewer 2 from the authors. In fact, indeed, there was a severe inadequate point on the presentation of Figure 4. Why should I evaluate this as “All results are highly convincing, and the conclusion drawn are comprehensible.” to the question “Are the results clearly presented?”

Authors’ response: We did not mean to criticize reviewer 2. As ad hoc reviewers ourselves, we would never expect different reviewers to have identical comments. We just meant that we were surprised that the reviewers’ comments were so drastically opposed to one another. Also, we did not understand why “Not applicable” was checked, which implies that no results were presented. If the reviewer thought that the results were not clearly presented, we would have expected “Can be improve” or “Must be improved” would have been selected.

There are several illogical points in the answer to the points 1 and 2.

  (1) The authors do not appreciate the additional assessment by using positive and/or negative control peptides suggested by reviewer 2. Then, how they validate and conclude that CD200AR really works in their vaccination system? Additionally, even though they do not agree to the suggestion, they did that “dispensable” experiments in rodents, and stand on this previously executed study as one of the reason to refuse the suggestion from reviewer 2. Why the study should be done on only rodents? Why the study is dispensable on canine? Do they have any evidence that CD200 action is same between rodents and canine?

(2) As rebuttal to the study using positive and/or negative control, they point an ethical reason. Then, how they explain the pertinence of the placebo experiment in clinical research in human patients.

(3) What is the ethical reason why placebo experiments can be done on human but control experiments can not be done on canine? Similarly, what is the ethical reason why control experiments can be done on rodents but not in canine?

Authors’ response: These statements respond to all of the reviewer’s comments. We do appreciate the use of appropriate positive and negative controls when performing experiments. We have been using the translational medicine paradigm that supports doing initial testing in induced murine models to test novel therapies, elucidating their mechanism of action if possible as well as assessing safety and efficacy. In the past, novel cancer therapies that were shown to be effective in murine models, some to the point of curing mice, were then moved directly to human clinical trials. However, this treatment development paradigm has been called into question, since It has been published that less than 11% of oncology drugs that work in mice are ever approved for human use. (Holzapfel BM, Wagner F, Thibaudeau L, Levesque JP, Hutmacher DW. Concise review: humanized models of tumor immunology in the 21st century: convergence of cancer research and tissue engineering. Stem Cells. 2015;33: 1696-1704; Ciociola A, Cohen L, Kulkarni P, et al. How drugs are developed and approved by the FDA: current process and future directions. Am J Gastroenterol. 2014;109: 620-622.) Using pet dogs with naturally occurring tumors for preclinical trials has been promoted as an intermediate step before advancing therapies to human trials. That is the research model that we have been using for several years. We have done our due diligence testing the peptides in our murine glioma model using them with and without lysate vaccines, and using scrambled peptides as well. The peptide is designed to act on the CD200 immune checkpoint. As with all checkpoint inhibitors, they can work alone with highly immunogenic tumor, such as melanoma; CNS tumors are not very immunogenic requiring the simultaneous use of tumor antigens to elicit an anti-glioma response. We have shown in the murine model that the peptide injected outside the CNS without lysate vaccine was ineffective in producing a positive response due to the lack of available antigens. Moreover, in both murine and in vitro human studies, we used scrambled peptides as controls with no demonstrable effect confirming the positive responses we’ve seen are due to the peptide. We saw that the peptide enhanced the efficacy of the lysate vaccines in mice before knowing the mechanism of action and decided to test it pet dogs with spontaneous high-grade glioma, which is what we present in the manuscript. We do not believe that something that has been shown to be ineffective in a murine model would be tested in a human clinical trial patient and think the same should apply to canine clinical trial patients – yes, we felt that would be unethical.

3. Good discussion. Hope good results for the current work.

Authors’ response: Thank you. We have moved the results of this pilot study to an NIH-supported canine clinical trial and are also moving toward Phase I/II human clinical trials.

4. To whom the apology from the authors was made? If to the reviewer 2, the reviewer will willing to accept the apology, however, are there any scientific meanings/significance? Should the apology be made to a reviewer, and be included in scientific comments? I would like to ask the decision of the editor.

Authors’ response: The apology was to reviewer 2 and the editor. The data presented on the effect of CD200 on human GBM patients helped support our decision to test the CD200 peptide in dogs.

 Round  3

Reviewer 2 Report

The author says they were surprised to have opposing opinions from 2 reviewers. However, I think such evaluation toward reviewer's comment is not suitable in the scientific reply from the author, according to the reason below. If the author want to evaluate the reviewers, at least the evaluation should be evenly made to the both favorable and oppositional. Unless doing so, authors should be aware of that such kind of thing has a risk to be understood as affective but not scientific. 

For me, presentation of any figure without any mention is "not applicable", and I simply evaluated so on the corresponding part. However, it was not toward the total manuscript, and I asked revision. In this 2nd reply, the author is still evaluating the reviewer2. 

Again I would like to say, I want logical discussion only. 

To each point

Points 1 and 2

(1) The description about the advantage of using canine as subject in the Introduction section (line 51-60), should be supported by some citations. If not, clear declaration that the description is authors' view should be added.

(2) The results of the experiments using negative control peptide are common between murine (in vivo) and human (in vitro) can be one of the good material to support the experimental design omitting that peptide in canine case. Is this fact clearly described in the manuscript? I may miss it, but if not, this description should be added.

(3) As the author pointed, now all we know it is so risky to apply knowledge from murine directly to human. Reviewer2 highly appreciate the points of focusing canine as problem-solving subject as well as spontaneous GBM, in current manuscript. Thus, I intended to accept the 2nd version of the manuscript if specific reasons based on some facts on pet dogs or spontaneous GBM (for example, rareness of GBM in dog, limitation of the sample numbers the authors could prepare, the meaning of "owner" that is quite different from human patients, and so on) which limit the experimental design was presented, and suggested additional experiments in the 1st comment as "unless experimentally impossible". However, I guess this intention was not reached to the authors from the 1st reply that insists ethical reason only. I judged that was not enough, and made 2nd comment. 

Unless some reason listed in (3) as example for reasons that limit the experimental design is not presented addition to above (2), readers will have strange feeling on this paper.

Point 4

Again I would like to say I think that being logical is the only important in scientific discussions. Since current conversation is the scientific discussion on the way aimed at publication, apology is not required and is even not useful.

I think apology is only required toward the scientific society and readers upon erratum or retraction.  

Author Response

Comments and Suggestions for Authors

The author says they were surprised to have opposing opinions from 2 reviewers. However, I think such evaluation toward reviewer's comment is not suitable in the scientific reply from the author, according to the reason below. If the author want to evaluate the reviewers, at least the evaluation should be evenly made to the both favorable and oppositional. Unless doing so, authors should be aware of that such kind of thing has a risk to be understood as affective but not scientific. 

For me, presentation of any figure without any mention is "not applicable", and I simply evaluated so on the corresponding part. However, it was not toward the total manuscript, and I asked revision. In this 2nd reply, the author is still evaluating the reviewer2. 

Again I would like to say, I want logical discussion only.

 Authors’ response: Thank you.

 To each point

Points 1 and 2

(1) The description about the advantage of using canine as subject in the Introduction section (line 51-60), should be supported by some citations. If not, clear declaration that the description is authors' view should be added.

 Authors’ response: We have added references about the advantages of using pet dogs with spontaneous cancers including glioma.

 (2) The results of the experiments using negative control peptide are common between murine (in vivo) and human (in vitro) can be one of the good material to support the experimental design omitting that peptide in canine case. Is this fact clearly described in the manuscript? I may miss it, but if not, this description should be added.

 Authors’ response: We added information describing the use of scrambled peptide as a negative control in both in vitro human cells and in vivo murine models.

 (3) As the author pointed, now all we know it is so risky to apply knowledge from murine directly to human. Reviewer2 highly appreciate the points of focusing canine as problem-solving subject as well as spontaneous GBM, in current manuscript. Thus, I intended to accept the 2nd version of the manuscript if specific reasons based on some facts on pet dogs or spontaneous GBM (for example, rareness of GBM in dog, limitation of the sample numbers the authors could prepare, the meaning of "owner" that is quite different from human patients, and so on) which limit the experimental design was presented, and suggested additional experiments in the 1st comment as "unless experimentally impossible". However, I guess this intention was not reached to the authors from the 1st reply that insists ethical reason only. I judged that was not enough, and made 2nd comment. 

Unless some reason listed in (3) as example for reasons that limit the experimental design is not presented addition to above (2), readers will have strange feeling on this paper.

 Authors’ response: The authors respectfully disagree with the reviewer that further reasons for not including a CD200 only treatment group in pet dogs with glioma would not be understood and accepted by the majority of readers. This translational “model” is well understood and accepted in cancer research including neuro-oncology.

 Point 4

Again I would like to say I think that being logical is the only important in scientific discussions. Since current conversation is the scientific discussion on the way aimed at publication, apology is not required and is even not useful.

I think apology is only required toward the scientific society and readers upon erratum or retraction.  

 Authors’ response: Understood.